# Inhibition of Ferroptosis Enables Safe Rewarming of HEK293 Cells following Cooling in University of Wisconsin Cold Storage Solution

**DOI:** 10.3390/ijms241310939

**Published:** 2023-06-30

**Authors:** Lucas P. Gartzke, Koen D. W. Hendriks, Femke Hoogstra-Berends, Christian P. Joschko, Anne-Lise Strandmoe, Pieter C. Vogelaar, Guido Krenning, Robert H. Henning

**Affiliations:** 1Department of Clinical Pharmacy and Pharmacology, University Medical Centre Groningen, University of Groningen, 9713 GZ Groningen, The Netherlands; l.p.gartzke@umcg.nl (L.P.G.);; 2Sulfateq B.V. Admiraal de Ruyterlaan 5, 9726 GN Groningen, The Netherlands

**Keywords:** 6-chromanols, SUL150, Ferrostatin-1, transplantation, hypothermia, organ preservation, lipid peroxidation, ferroptosis, HEK293

## Abstract

The prolonged cooling of cells results in cell death, in which both apoptosis and ferroptosis have been implicated. Preservation solutions such as the University of Wisconsin Cold Storage Solution (UW) encompass approaches addressing both. The use of UW improves survival and thus extends preservation limits, yet it remains unclear how exactly organ preservation solutions exert their cold protection. Thus, we explored cooling effects on lipid peroxidation and adenosine triphosphate (ATP) levels and the actions of blockers of apoptosis and ferroptosis, and of compounds enhancing mitochondrial function. Cooling and rewarming experiments were performed in a cellular transplantation model using Human Embryonic Kidney (HEK) 293 cells. Cell viability was assessed by neutral red assay. Lipid peroxidation levels were measured by Western blot against 4-Hydroxy-Nonenal (4HNE) and the determination of Malondialdehyde (MDA). ATP was measured by luciferase assay. Cooling beyond 5 h in Dulbecco’s Modified Eagle Medium (DMEM) induced complete cell death in HEK293, whereas cooling in UW preserved ~60% of the cells, with a gradual decline afterwards. Cooling-induced cell death was not precluded by inhibiting apoptosis. In contrast, the blocking of ferroptosis by Ferrostatin-1 or maintaining of mitochondrial function by the 6-chromanol SUL150 completely inhibited cell death both in DMEM- and UW-cooled cells. Cooling for 24 h in UW followed by rewarming for 15 min induced a ~50% increase in MDA, while concomitantly lowering ATP by >90%. Treatment with SUL150 of cooled and rewarmed HEK293 effectively precluded the increase in MDA and preserved normal ATP in both DMEM- and UW-cooled cells. Likewise, treatment with Ferrostatin-1 blocked the MDA increase and preserved the ATP of rewarmed UW HEK293 cells. Cooling-induced HEK293 cell death from hypothermia and/or rewarming was caused by ferroptosis rather than apoptosis. UW slowed down ferroptosis during hypothermia, but lipid peroxidation and ATP depletion rapidly ensued upon rewarming, ultimately resulting in complete cell death. Treatment throughout UW cooling with small-molecule Ferrostatin-1 or the 6-chromanol SUL150 effectively prevented ferroptosis, maintained ATP, and limited lipid peroxidation in UW-cooled cells. Counteracting ferroptosis during cooling in UW-based preservation solutions may provide a simple method to improve graft survival following cold static cooling.

## 1. Introduction

In clinical kidney transplant procedures, simple cold static hypothermic (4 °C) preservation remains the most widely employed strategy to avoid ischemia reperfusion injury (IRI) [1] and ensure graft health [2]. During this procedure, organs are cold-flushed with and stored in ice-cooled organ preservation solutions such as the University of Wisconsin Cold Storage Solution (UW) or an alternative organ preservation solution (Custidiol, HTK, etc.). The traditional perspective is that this forced hypothermia protects the graft from IRI by exogenously lowering metabolism and limiting oxidative stress, thereby maintaining energy reserves and cell viability [2,3]. However, cooling efficacy is limited, as it does not fully prevent the formation of reactive oxygen species (ROS) and graft damage [1,2,3,4,5,6,7,8,9,10].

Previously, we showed cooling to decrease mitochondrial respiration according to the Q10 effect, which describes the reaction rate changes of a biochemical reaction per 10 °C temperature change. Specifically, cooling at 4 °C resulted in a ~40% (2 h) and 90% (16 h) inhibition of ATP formation, yet lowered ROS production only by half [11]. Additionally, hypothermia increased the ROS burden by impairing ROS-scavenging enzymes such as superoxide dismutase. In keeping with these findings, we showed that 2 h of cooling doubles the levels of lipid peroxidation [12]. Consequently, cooling slows down, but does not fully preclude, the damaging processes. Indeed, although the recommended limit of kidney hypothermia is 18 h [5], graft survival decreases dramatically from 6 h of preservation time onwards [7].

We and others showed earlier that isolated cells in standard media die from or during hypothermia [11,12,13,14,15], although the exact mechanisms are still debated. One of the suggested pathways is (intrinsic) apoptosis triggered by excess calcium influx in response to cell membrane depolarization because of the failure of the sodium–potassium pump due to a hypothermia-induced loss of ATP [15,16,17,18,19]. This suggestion is supported by the prevention of cold-mediated cell death by calcium chelators and calcium channel blockers [15,17]. While calcium chelators such as EDTA, BAPTA-AM, and dopamine protect against hypothermia, they do not exclusively chelate calcium and show functional overlap with other chelators of cations (such as ferrous iron). Therefore, their protective action cannot easily be attributed to a single ion [17,20], which has challenged the understanding of the role of calcium in hypothermic cell death and implicated the involvement of iron in inducing ferroptosis [21,22,23,24].

Ferroptosis is uniquely characterized among cell death pathways by its iron-dependent and ROS-mediated accumulation of lipid peroxidation and subsequent depletion of plasma membrane polyunsaturated fatty acids [25,26,27,28,29,30,31,32,33] without observable nuclear changes (such as chromatin condensation) [34,35]. Ferroptosis is counteracted by the enzyme family of glutathione peroxidases (GPxs), which reduce glutathione (GSH) to neutralize ROS. Specifically, the isoform GPx4 reduces membrane-bound oxidized lipids and serves as the specific endogenous cellular defense against ferroptosis [35,36,37,38,39]. Lack of GPx4 under oxidative conditions promotes cell death via ferroptosis [34,40]. In keeping with this, the targeted depletion of cellular glutathione (GSH) reservoirs or direct inhibition of GPx4 result in the induction of ferroptosis via the increased formation of lipid peroxides [34,40]. Conversely, the reduction of the available intracellular iron pool with deferoxamine protects cells from ferroptosis during in vitro normothermia and—more importantly—during in vivo hypothermic organ preservation [17,21,22,23,24]. Ferrous iron (Fe^2+^) is central in the generation of lipid peroxides, as it initiates the Fenton reaction in the presence of (increased) mitochondrial hydrogen peroxide (H_2_O_2_), generating ferric iron (Fe^3+^), HO• (hydroxide radical), and OH- (hydroxide ion) (Fe^2+^ + H_2_O_2_
**→** Fe^3+^ + HO^•^ + OH^−^). This conversion of H_2_O_2_ into the more reactive HO^•^, in turn, causes secondary oxidative reactions, among them the oxidation of membrane polyunsaturated fatty acids (PUFAs). 

Ferroptosis can be prevented by direct inhibitors of lipid peroxidation, such as Ferrostatin-1, a synthetic antioxidant that scavenges alkoxyl radicals and is regenerated by ferrous iron and, therefore, is not consumed while executing its scavenging properties [41]. A proposed role of ferroptosis is substantiated by our finding that Ferrostatin-1 attenuates hypothermic tissue damage and cell death [14,42]. ROS overload itself, as well as the damage that ensues from it, such as DNA damage or lipid peroxidation, can trigger (intrinsic) apoptosis and ferroptosis, respectively. Whereas ferroptosis, unlike other forms of regulated cell death, such as apoptosis, is strictly dependent on ROS-mediated lipid peroxidation [25,26,27,30,31,32,33], compounds that limit excessive ROS accumulation may protect cells from cell death via both pathways. Consequently, the limitation of mitochondrial ROS production is another approach to limit ferroptosis and explains the cold tolerance induced by 6-chromanols [13,16,43,44,45]. These compounds are structurally related to both dopamine and vitamin E, and stabilize mitochondrial function by maintaining complex I and IV functions, resulting in stable ATP production under hypothermic conditions [43,44]. In rats, 6-chromanols maintain structural kidney morphology after prolonged total body hypothermia [43].

In clinical practice, the use of UW has improved outcomes and thus extended safe preservation limits, yet it remains unclear how exactly organ preservation solutions exert their cold protection [11,46,47]. Interestingly, UW encompasses approaches addressing both apoptosis and ferroptosis. UW features sodium and potassium concentrations matching those of the intracellular milieu [48], thus alleviating hypothermia-induced plasma membrane depolarization and subsequent excessive calcium influx and ensuing (intrinsic) apoptosis. Furthermore, UW is rich in GSH [49,50,51], which provides the antioxidative capacity to sustain GPx4 activity, with the potential to prevent ferroptosis [25,26,27,36,37,38,39,40]. Here, we aimed to establish the mechanism of cell death in a cellular model for renal cold storage in UW and explored the efficacy and timing of pharmacologic interventions with different antioxidants and 6-chromanols.

## 2. Results

### 2.1. Hypothermia Causes Cell Death in HEK293 Cells That Is Partially Ameliorated by UW

As an initial step, we determined cell viability by neutral red assay after cooling for 5, 10, and 24 h in both regular medium (Dulbecco’s Modified Eagle Medium, DMEM) and University of Wisconsin Cold Storage Solution (UW). In DMEM, 5 h of cooling did not cause HEK293 cell death (Figure 1b), whereas viable cells were absent after 10 or 24 h of cooling. In contrast, 5 h of cooling in UW lowered cell survival by ~40%, after which cell survival gradually decreased at 10 and 24 h of cooling (Figure 1c).

### 2.2. Inhibition of Apoptosis Does Not Increase Cell Survival

To investigate whether cooled HEK293 cells die of apoptosis, we first tested cellular and mitochondrial calcium channel blockades’ prevention of cell death during hypothermia and after rewarming. A broad-range blockade of extracellular Ca^2+^ channels by lanthanum chloride (50 mM, Figure 2a,b) and the mitochondrial Ca^2+^ uniporter by Ru360 (5 mM, Figure 2c,d) did not prevent cell death during 24 h cooling in DMEM or UW (Figure 2a–d). Next, we explored apoptosis inhibition by Z-VAD(OH)-FMK, a pan-caspase inhibitor with a specific affinity for the final apoptotic effector caspase 6 and a preserved function in the cold (Figure A3). Like the Ca^2+^ blockade, the inhibition of caspase 6 with Z-VAD(OH)-FMK (50 µM, Figure 2e,f) did not prevent cell death in DMEM-cooled cells, and only very partly rescued UW-cooled cells from cell death following rewarming. Collectively, these results show that hypothermia-induced HEK293 cell death cannot be prevented by inhibiting apoptosis.

### 2.3. Addition of SUL150 and Ferrostatin-1 Preventing Ferroptosis

Next, we investigated whether the 6-chromanol SUL-150, which preserves mitochondrial function [13,16,43,44,45], could also attenuate ferroptosis, which it did (Figure A4). We tested the efficacy of SUL150 and Ferrostatin-1, a known inhibitor of lipid peroxidation and ferroptosis, to exert protection from cooling and/or from cooling and rewarming. DMEM or UW were supplemented with SUL150 (10 µM) or Ferrostatin-1 (100 nM) at the start (0 h) of 24 h of cooling. Both SUL150 and Ferrostatin-1 completely inhibited cell death in either DMEM- or UW-incubated cells both at the end of cooling (Figure 3) and following rewarming (Figure 4).

### 2.4. Rewarming Causes ATP Depletion and Lipid Peroxidation in UW Cold-Preserved HEK293 Cells

Paradoxically, whereas preservation in UW conferred robust cell survival during cooling, as shown by light microscopy and neutral red survival assay, the rewarming of cells resulted in imminent cell death. To explore the mechanism of the cell death of cooled UW cells upon rewarming, we measured lipid peroxidation by MDA assay and the ATP content by luciferase assay. The cooling of HEK293 for 24 h in UW followed by rewarming in DMEM for 15 min induced a ~50% increase in MDA (Figure 5b), while concomitantly lowering ATP by >90% (Figure 6b). Treatment with either SUL150 or Ferrostatin-1 effectively precluded the increase in MDA and decrease in ATP seen in rewarmed UW-cooled cells (Figure 5 and Figure 6). It is of note that Ferrostatin-1 did not fully preserve the ATP content of cooled and rewarmed DMEM cells (Figure 6e). Vehicle-treated DMEM-incubated HEK293 cells could not be explored, as they do not survive without intervention. Thus, we conclude that cooling and rewarming increases lipid peroxidation and depletes ATP in UW-cooled cells upon rewarming, which is precluded by either SUL150 or Ferrostatin-1.

### 2.5. UW Prevents Oxidative Stress in the Cold

Given the increased lipid peroxidation in UW cooled-and-rewarmed cells, we next asked whether peroxidation is already present during hypothermia or depends on rewarming, and determined this based on an abundance of 4HNE, an obligatory and specific product of lipid peroxidation. 4HNE levels increased after 5 h of cooling of HEK293 cells, both in UW and DMEM (Figure 7b,c). However, in UW-cooled cells, 4HNE abundance returned to baseline at 10 and 24 h of cooling (Figure 7c), whereas DMEM cells did not tolerate cooling beyond 5 h. Treatment with SUL150 (10 µM) or Ferrostatin-1 (100 nM) prevented the increase in lipid peroxidation in cooled UW cells (Figure 7e,g). In DMEM-cooled cells, SUL150 (10 µM) delayed, and Ferrostatin-1 completely prevented, a 4HNE increase (Figure 7d,f). Thus, UW protects cells from increased lipid peroxidation during prolonged cooling, and the initial increase is blocked by SUL150 and Ferrostatin-1.

### 2.6. Antioxidant Supplementation Shortly before or during Rewarming Does Not Prevent Cell Death

Given the increased lipid peroxidation and, ultimately, cell death in UW-cooled cells following rewarming, we reasoned that UW treatment may provide initial cell protection because of its high glutathione (GSH) content (3 mM), which is lost upon the necessary replacement with DMEM upon rewarming (because of ionic composition). To test this, cell survival was assessed in HEK293 cells supplemented either 30 min before or immediately upon rewarming with oxidized (see Appendix A; Figure A1) and reduced GSH, the lipid-soluble antioxidant vitamin E (Trolox), the 6-chromanol SUL150, and Ferrostatin-1. Compound supplementation either before or during rewarming did not increase cell survival in UW-cooled HEK293 cells (Figure 8 and Figure 9).

### 2.7. UW Does Not Preclude Progressive Loss of GPx4

Due to the lack of effect of the compounds with administration shortly before or at the time of the initiation of rewarming, we next asked whether cooling affects GPx4 abundance, and measured its protein expression in all conditions and at all time points. The cooling of HEK293 in DMEM for 5 h significantly reduced GPx4 protein levels (Figure 10b), with cooling for 10 or 24 h resulting in complete cell death. In UW-cooled cells, GPx4 levels were unaffected after 5 and 10 h of hypothermia, but significantly decreased at 24 h (Figure 10c).

Next, we investigated whether levels of GPx4 were maintained through supplementation of DMEM and UW with SUL150 (10 µM). In DMEM, SUL150 supplementation resulted in stable GPx4 levels following 5 h cooling, but its abundance reduced significantly at 10 and 24 h (Figure 10d). The addition of Ferrostatin-1 (100 nM) to DMEM maintained GPx4 levels throughout the experiment (Figure 10f). In contrast, SUL150 treatment of UW-cooled cells significantly increased GPx4 after 5 h cooling, which returned to baseline level following 10 and 24 h cooling (Figure 10e). Cooling in UW with Ferrostatin-1 did not result in significant changes in GPx4 expression; however, we observed an increase after 5 h of cooling, which did not reach significance (Figure 10f,g). Thus, the time-dependent loss of endogenous antioxidative defenses against ferroptosis can be prevented by SUL150 and Ferrostatin-1 in UW-cooled HEK293 cells.

### 2.8. Proteasome Inhibition Does Not Prevent GPx4 Loss or Cell Death

Because we observed a reduction of GPx4 levels during hypothermia, both in DMEM and UW, we next investigated whether this was due to proteasomal breakdown. The proteasome inhibitor, MG132 (10 mM), did not increase survival during cooling or rewarming (Figure 11b). It should be noted that MG132 was well tolerated in normothermic cells at 1 h of treatment, yet induced complete cell death after 24 h (Figure 11a).

## 3. Discussion

The combination of hypothermia and University of Wisconsin Cold Storage Solution (UW) has improved organ preservation outcomes [49,50,51]. While the mechanism of hypothermia, as well as its benefits and drawbacks, are well documented in the literature [1,2,3,4,5,6,7,8,9,10,11,46,47], the protective mechanism of UW has not been fully elucidated yet. Here, we show that UW indeed protects from hypothermic cell death. However, upon rewarming, cell death is executed through ferroptosis, driven by ATP depletion and lipid peroxidation. Finally, treatments that maintain mitochondrial health or inhibit lipid peroxidation throughout cooling counteract ferroptosis and rescue UW-cooled cells from rewarming-induced cell death.

### 3.1. UW-Cooled Cell Death Is Executed through Ferroptosis Rather Than Apoptosis

We identified ferroptosis as the pathway conferring cell death following rewarming of UW-cooled cells. Small-molecule Ferrostatin-1, an inhibitor of lipid peroxidation, can prevent ferroptosis. Here and earlier [14], we have demonstrated that Ferrostatin-1 can confer protection from hypothermic cell death. We showed that cooling in DMEM causes rapid and complete cell death by ferroptosis. UW extends cooling time significantly, but rewarming results in massive cell death, likely due to increased lipid peroxidation and ATP depletion. Ferroptosis is caused by lipid peroxidation in the absence of a sufficient antioxidative defense system, i.e., glutathione peroxidase 4 (GPx4), which reduces glutathione (GSH) and thereby counteracts lipid peroxidation [25,26,27,34,35,36,37,38,39,40]. We demonstrated that hypothermia causes a reduction in GPx4 protein abundance in HEK293 cells after 5 h of cooling in DMEM-cooled cells, accompanied by a concomitant increase in lipid peroxidation levels. UW delayed, but did not fully prevent, a cooling-evoked time-dependent decrease in GPx4; i.e., GPx4 levels were maintained at 5 and 10 h of cooling, but were significantly reduced following 24 h of cooling. In addition, UW-cooled HEK293 cells also showed increased lipid peroxidation at 5 h of hypothermia. We conclude that UW alone substantially precludes ferroptosis during hypothermia, most likely due to its high GSH content fueling the activity of Gpx4, thus counteracting the early increase in lipid peroxidation, and effectively extending cell survival.

### 3.2. ATP Depletion, Calcium, and Hypothermic Cell Death

We and others have shown that during hypothermia, ATP levels are depleted [11,13,14,17,18,44]. Here, we demonstrated ATP depletion in 24 h UW-cooled cells during rewarming. It has been proposed that ATP depletion causes cessation of the sodium–potassium pump, which leads to membrane depolarization, voltage-gated calcium influx, and finally apoptosis [15,16,17,18]. While we did show ATP depletion, our data indicate that cells do not die of apoptosis during cooling and rewarming, as solidified both by the ineffectiveness of blocking a calcium influx and by inhibiting the final effector of apoptosis by a pan-caspase inhibitor. Importantly, pharmacologic interventions that allowed for safe cooling and rewarming (SUL150; Ferrostatin-1) maintained ATP levels in UW-cooled cells, but also prevented lipid peroxidation. Conversely, Bayley et al. [15] showed that hypothermic cell death, which they attributed to apoptosis, was prevented by a blockade of voltage-gated calcium channels (lanthanum) in insect muscle tissue, and others have demonstrated the effectiveness of calcium chelation [17]. It is possible that mammalian hypothermic cell death is regulated differently from insect cell death; however, recent literature has shown that ferroptosis is evolutionarily conserved, and also takes place in insects [52]. This corresponds with ferroptosis’ independence from transcriptional activity, as it is the effect of a disturbed intracellular microenvironment—the redox balance of the cell [35,41]. In keeping with this, calcium is an important regulator of mitochondrial activity, and an increase in mitochondrial calcium concentrations, i.e., a mitochondrial matrix calcium overload, can cause mitochondrial dysfunction, along with lowered ATP and excessive ROS production [18], the latter of which is central in ferroptosis [25,26,27,28,35]. Advanced preservation techniques, such as pressure-controlled oxygenated machine perfusion, create fewer complications, reduced oxidative stress, and increased energy levels [53,54,55]. An attractive hypothesis is that these techniques maintain or restart mitochondrial respiration through oxygen delivery, and thereby maintain the ATP [53] to support antioxidant transport from the preservation solution, which fails in a static cold storage solution due to progressive ATP depletion. In doing so, oxygenated machine perfusion may prevent mitochondrial dysfunction, maintaining sufficient ATP and antioxidant delivery to prevent ferroptosis by precluding excessive accumulation of lipid peroxides upon rewarming.

### 3.3. Early Treatment Prevents Ferroptosis

Early treatment with either SUL150 or Ferrostatin-1, i.e., at the start of cooling, proved effective in counteracting increased lipid peroxidation during rewarming, while concomitantly maintaining ATP levels. Importantly, the addition of these compounds and other antioxidants, including glutathione (GSH) and Trolox, at the end of cooling or during rewarming did not increase cell survival. One reason for this could be that GSH is not taken up by the cells, i.e., cannot pass the lipid phase of the cell membrane [56], as it may only be taken up in its ester form [57,58]. However, research in cold-preserved rodent livers has demonstrated the effectiveness of supplying extra GSH during rewarming [59]. Uptake alone, however, does not suffice to explain the failure associated with late administration of these compounds. The water-soluble antioxidant Trolox did not reduce cell death when administered shortly before or during rewarming, although it passes the cellular membrane [60] and prevents ferroptosis [25]. One reason for this could be a failure of antioxidant delivery in the context of ATP depletion. GSH requires active transport via ATPase [56,59], whereas Trolox passes the cell membrane; however, the exact mechanism of action has yet not been elucidated. Alternatively, the loss of GPx4, i.e., the ferroptosis-specific oxidative defense, could explain the ineffectiveness of late intervention. Hypothermia-dependent GPx4 loss is not an active cellular process caused by ubiquitination, as levels of ubiquitin did not differ over time or treatment. Furthermore, cooling with a proteasome inhibitor did not increase cell survival following cooling or rewarming. Oxidative stress itself provides an alternative to the loss of GPx4. Due to its nine cysteines, which can be readily oxidized, GPx4 itself is prone to oxidative damage via disulfide bridge formation, which, in turn, causes confirmation changes [61,62] and may result in protein breakdown. This GPx4 loss may explain why the addition of exogenous GSH during or shortly before rewarming did not prevent cell death, as supplying the substrate GSH in the absence of sufficient GPx4 is ineffective. Alternatively, ATP depletion may inhibit both GSH uptake into the cell and/or the intracellular transport and processing of either GSH or Trolox, and, therefore, fail to exert protection upon rewarming. Finally, the increased effectivity of Ferrostatin-1 compared to other antioxidants may lie in the chain-breaking and iron-scavenging properties of Ferrostatin-1. Trolox is indeed a water-soluble antioxidant that confers its protection through the donation of a hydrogen atom to the lipid peroxyl radicals, thereby preventing their accumulation. Conversely, Ferrostatin-1 scavenges the alkoxyl radicals which are formed from hydroperoxides by ferrous iron (Fe^2+^), and, additionally, binds ferrous iron from the cytosol which would otherwise initiate lipid peroxidation—a reaction known as the Fenton Reaction. Furthermore, Ferrostatin-1 is self-recycling, as it is regenerated but ferrous iron, which reduces the Ferrostatin-1-radical complex. In lipid peroxidation induced through iron, Ferrostatin-1 has far greater chain-breaking capacity than Trolox [41], which explains Ferrostatin-1’s superiority over Trolox, as was also shown by us in earlier work.

### 3.4. Limitations

While human HEK293 cells are extensively used in research, we should note that the cells are of tumorigenic origin. Although we do not have reason to believe that their response to cold differs from primary kidney cells, we did not prove this.

## 4. Materials and Methods

### 4.1. Cellular Cooling and Rewarming Model

Human Embryonic Kidney (HEK293, ATCC CRL-1573) cells were cultured in a humidified incubator with 20% oxygen, 5% carbon dioxide at 37 °C in Dulbecco’s Modified Eagle Medium (DMEM) supplemented with 10% fetal bovine serum and 1% penicillin-streptomycin. For cooling and rewarming experiments, six-well plates were coated with 0.03% poly-l-lysine and cells were grown to 90–95% confluency.

To perform hypothermia and rewarming experiments, cells were seeded in six-well plates and after overnight adherence placed at 4 °C in a standard laboratory refrigerator for different time periods, with or without rewarming by reinstitution of standard cell culture conditions (DMEM at 37 °C). Prior to cooling, cell culture plates were placed in open zip-locked bags to maintain the appropriate atmosphere. Subsequently, zip-locks were sealed, and cells were cooled in either DMEM or University of Wisconsin Cold Storage Solution (UW) in the absence or presence of compounds listed in Table 1. To mimic the clinical practices of organ preservation, including rapid temperature changes, a quick wash with 18 °C PBS preceded the final storage in cooling and rewarming media preconditioned at temperatures of 4 °C or 37 °C, respectively. Cells were observed with a Leica DMIL 090-135.002 microscope mounted with a Leica MC120 HD camera at 10× magnification (Leica 506271, 345IH/01). The pictures shown are representative images.

### 4.2. Neutral Red Cell Survival Assay

After removal of medium, cells were washed once with PBS, neutral red (NR) medium (50% DMEM; 50% HBSS (Hanks’ Balanced Salt Solution); 50 mg/mL NR dye) was added, and cells were incubated for 60 min at 37 °C followed by washing with PBS, lysis by NResorb solution (49% demineralized water; 50% ethanol 96% and 1% glacial acid), and incubation in the dark for 10 min on a shaker at room temperature. NR absorbance was measured at 540 nm using a Synergy 2 Multi-Mode plate reader. Data are expressed as relative absorbance with controls set to 1; a one-way ANOVA with Šidák’s or Dunnett’s post hoc test was used for statistical analysis.

Because non-viable HEK293 cells detach from the surface of the well, trypan blue assay for staining of dead cells could not be reliably performed. To ensure adhesion of viable cells, coating of well surfaces with 0.03% poly-l-lysine was performed prior to seeding. Poly-l-lysine 0.03% or Gelatin 1–2%, unfortunately, did not enable trypan blue assay for cell death analysis.

### 4.3. ATP Measurement

ATP was measured by a luciferase assay (Promega). Cells were collected after the addition of an EDTA buffer and scraping on ice. We employed 6 min boiling (95 °C) to lyse the cells and to denature proteins to stop ATP consumption [63]. Then, samples were centrifugated at 11,000 g for 2 min, and supernatant was collected. ATP was measured at 590 nm using a dark plate in a Synergy 2 Multi-Mode plate reader. Bio-Rad DC protein assay was performed to determine protein concentration, for which all read-outs were corrected. Data are expressed as relative luminescence corrected for sample protein concentration with controls set to 1; a one-way ANOVA with Dunnett’s post hoc test was used for statistical analysis.

### 4.4. MDA Measurement

Cells were harvested with PBS with 1% BHT (Butylhydroxytoluol) and immediately placed on ice. Cells were lysed with SDS for 5 min at room temperature and then boiled for 60 min with TBA reagent. MDA was measured at 590 nm using a dark plate in a Synergy 2 Multi-Mode plate reader. Bio-Rad DC protein assay was performed to determine protein concentration, for which all read-outs were corrected. Data are expressed as relative luminescence corrected for sample protein concentration with controls set to 1; a one-way ANOVA with Dunnett’s post hoc test was used for statistical analysis.

### 4.5. Western Blot

Cell lysates were obtained using RIPA lysis buffer (50 mM Tris-Cl pH = 8.0, 150 mM NaCl, 1% Igepal Ca 630, Sodium Deoxycholate, 1.0% SDS, 0.4% protein inhibitor cocktail, 1 mM sodium orthovanadate, 10 mM NaF). Additionally, cells were mechanically lysed through repetitive aspiration through a 30 G needle. Loading buffer (10% SDS, 50% Glycerol, 0.33 M Tris HCl pH = 6.8, 10% beta-mercaptoethanol, 0.05% bromophenol blue) was added (20% of total volume) to all samples, followed by boiling for 5 min at 95 °C. Samples were loaded on 4–20% SDS precast gels (Bio-Rad TGX gels) and transferred to nitrocellulose membrane (Bio-Rad). Membranes were blocked (50 mL TBST, 2.5 g skim milk powder) for 60 min. Antibodies were diluted 1:1000 in TBST containing 3% BSA (bovine serum albumin). Primary antibodies were incubated overnight at 4 °C. Secondary antibodies were incubated at room temperature for 60 min. Membranes were visualized using SuperSignal (PerkinElmer) and ChemiDoc MP imaging system (Bio-Rad). For incubation with a different primary antibody, membranes were incubated with a stripping buffer (45 mL TBS, 5 mL 20% SDS, 350 µL beta-mercaptoethanol) for 30 min at room temperature. Stripping efficiency was tested with the ChemiDoc MP imaging system and SuperSignal. Antibodies used were anti-4-Hydroxynonenal (Abcam, ab46545), anti-Glutathione-Peroxidase-4 (Abcam, ab125066), anti-Mono- and polyubiquitinylated conjugates (Enzo, ENZ-ABS840-0500), GARPO (Dako), RAMPO (Dako). A master sample derived from pooling all samples was run along with intra- and intermembrane protein normalization and comparison. In ImageLab 6.0 (Bio-Rad), densitometric data was normalized using the stain-free normalization method. Data were expressed as relative protein abundance with normothermic controls set to 1. A one-way ANOVA with Dunnett’s post hoc test was used for statistical analysis.

### 4.6. Caspase 3/7 Glow Assay

To test the efficacy of Z-VAD(OH)-FMK, a pan-caspase inhibitor with specific affinity for the final apoptotic effector caspase 6, rat smooth muscle primary cells (SMAC) were cultured in sterile 96-well dark plates until confluent and incubated for 30 min at 37 °C and subsequently cooled to 4 °C for 30, 60, 90, 120, 150, and 180 min. Caspase activity was assessed using the Caspase-Glo^®^ 3/7 Assay (Promega, Madison, WI, USA) according to manufacturer’s protocol, and luminescence was measured using a Synergy 2 Multi-Mode plate reader (BioTek). Data were expressed as total luminescence absorbed. A two-way ANOVA with Tukey’s multiple comparisons test was used for statistical analysis.

### 4.7. Efficacy Assay: SUL150 versus Erastin and RSL3

To test the efficacy of SUL150 to directly counteract ferroptosis, we incubated HEK293 cells under normothermic conditions (37 °C) with the known ferroptosis inducers RSL3 (Ras-selective lethal 3) and Erastin with and without 10 µM of SUL150. RSL3 was incubated for 6 h in concentrations of 1.44 and 2.88 µM. Erastin was incubated for 24 h in concentrations of 5 and 10 µM. To assess cell viability and, by proxy, cell death a neutral red survival assay was performed according to the specifications described above. Data are expressed as relative absorbance with controls set to 1; a one-way ANOVA with Šidák’s or Dunnett’s post hoc test was used for statistical analysis.

## 5. Conclusions

In a cellular cooling and rewarming model for renal cold preservation, we showed that cell death from hypothermia and/or rewarming is caused by ferroptosis rather than apoptosis. We showed that University of Wisconsin Cold Storage Solution prevents ferroptosis during hypothermia, but lipid peroxidation and ATP depletion rapidly ensue upon rewarming, ultimately ending in complete cell death. We demonstrated that treatment with small-molecule Ferrostatin-1 or the 6-chromanol SUL150 with UW throughout cooling effectively prevents ferroptosis. Both compounds maintain ATP and limit lipid peroxidation. Because treatments are ineffective when administered shortly before or during rewarming, administration from the initiation of cooling onwards is necessary to maintain adequate ATP levels and counteract lipid peroxidation upon rewarming. Both SUL150 and Ferrostatin-1 might be efficacious, simple, and low-cost additions to the current organ preservation solutions to improve static or machine perfusion cold storage preservation tolerance of allografts and reduce complications such as ischemia reperfusion injury. Future studies should investigate the efficacy of targeting ferroptosis before, or concomitant with, the start of organ preservation protocols.

## Figures and Tables

**Figure 1 ijms-24-10939-f001:**
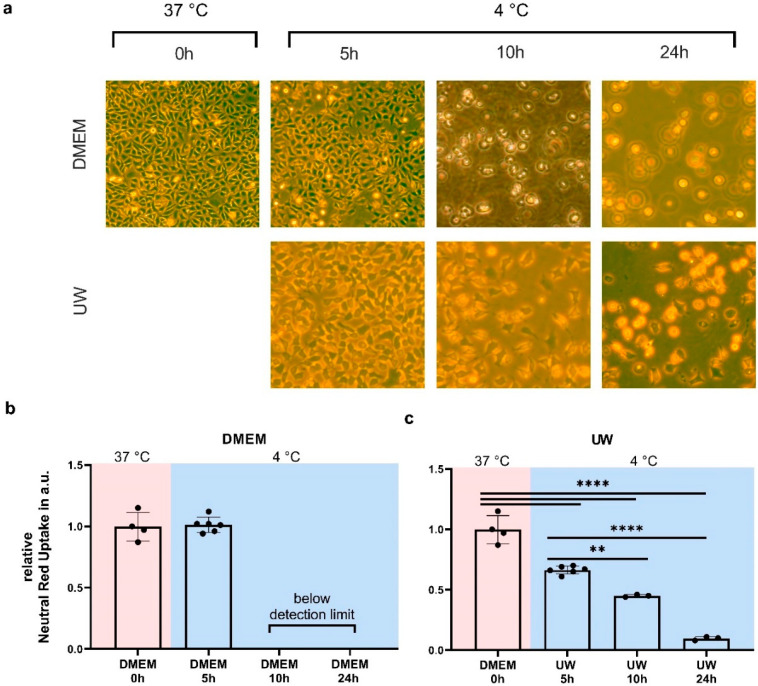
Cell survival in Human Embryonic Kidney (HEK) 293 cells after 5, 10, and 24 h of hypothermia (blue background) in cell medium (DMEM) or University of Wisconsin Cold Storage Solution (UW). (**a**) Typical microscopy pictures of conditions tested (10× magnification). (**b**) Cell survival after cooling in DMEM. (**c**) Cell survival after cooling in UW. All data are expressed relative to standard conditions at 37 °C and presented as means ± SD; ** = *p* < 0.005; **** = *p* < 0.0001; one-way ANOVA with Šidák’s post hoc test; *n* = 3–6 per condition.

**Figure 2 ijms-24-10939-f002:**
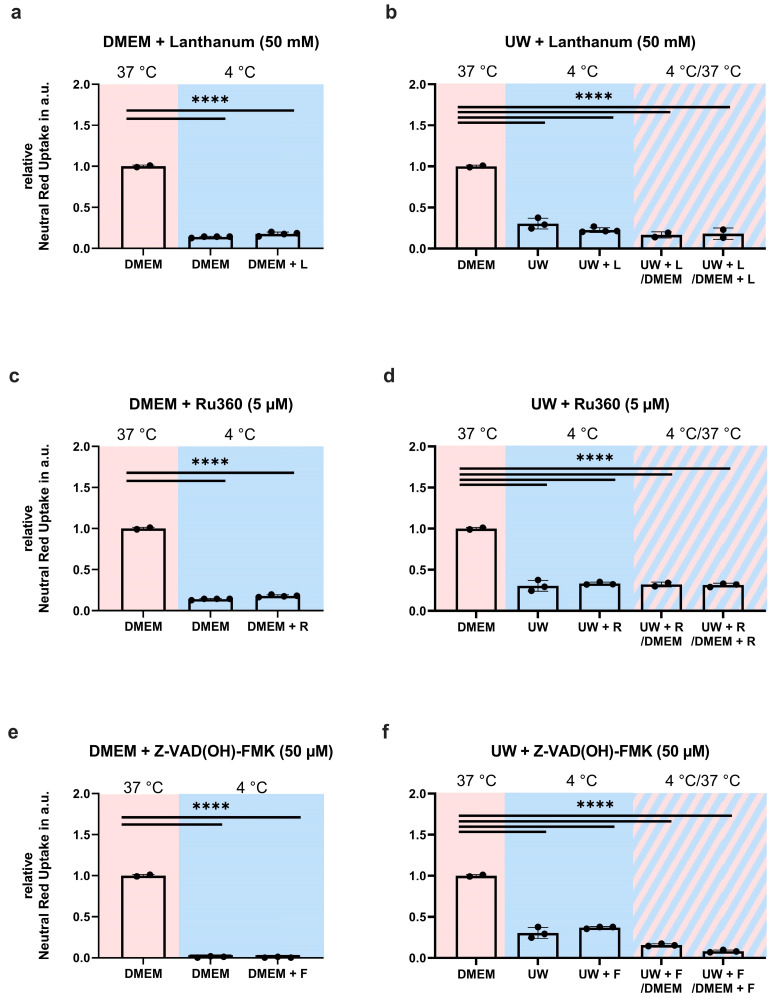
Cell survival in Human Embryonic Kidney (HEK) 293 cells following 24 h cooling (blue background) and 1 h rewarming (striped background) to test for apoptosis. (**a**,**b**) Broad-range blockade of extracellular Ca^2+^ channels by lanthanum chloride (L) in DMEM (**a**) and UW (**b**). (**c**,**d**) Blocking the mitochondrial Ca^2+^ uniporter with Ru360 (R) in DMEM (**c**) and UW (**d**). (**e**,**f**) Pan-caspase inhibition (F) with Z-VAD(OH)-FMK (F) in DMEM (**e**) and UW (**f**). All data are expressed relative to standard conditions at 37 °C and presented as means ± SD; one-way ANOVA with Šidák’s post hoc test; **** = *p* < 0.0001; *n* = 2–4.

**Figure 3 ijms-24-10939-f003:**
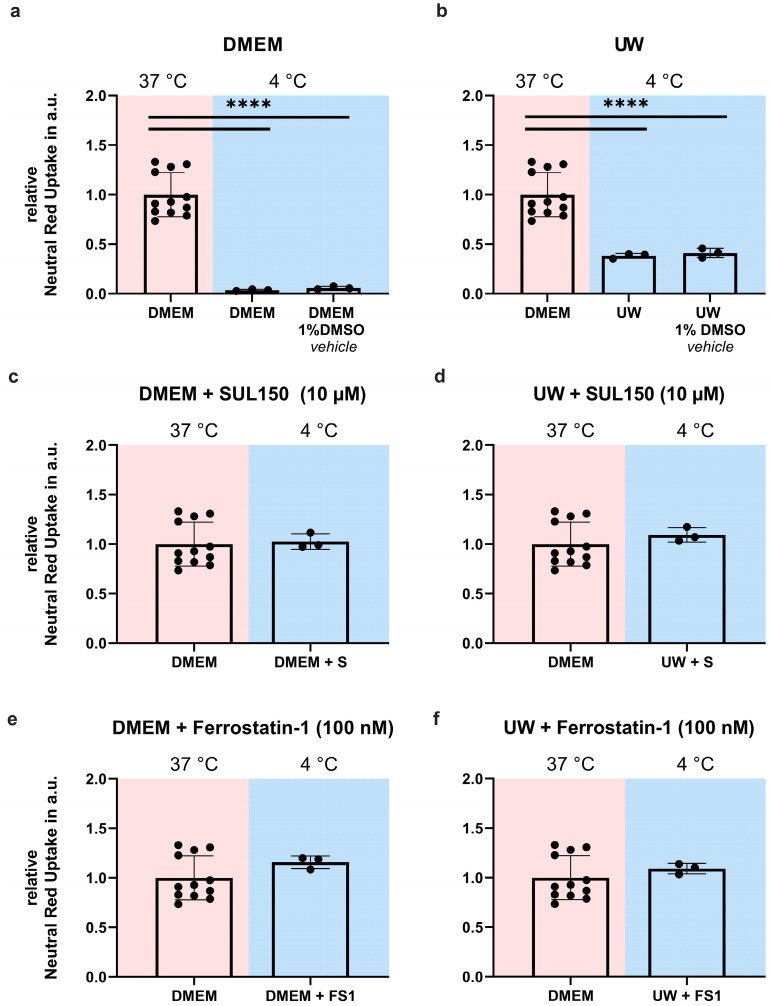
Cell Survival in Human Embryonic Kidney (HEK) 293 cells after 24 h of cooling (blue background) with SUL150 or Ferrostatin-1 supplementation in either DMEM or UW. (**a**,**b**) 24 h cooling with DMSO vehicle in DMEM (**a**) and UW (**b**). (**c**,**d**) 24 h cooling with SUL150 (S) in DMEM (**c**) and UW (**d**). (**e**,**f**) 24 h cooling with Ferrostatin-1 (F) in DMEM (**e**) and UW (**f**). All data are expressed relative to standard conditions at 37 °C and presented as means ± SD; **** = *p* < 0.0001; one-way ANOVA with Dunnett’s post hoc test; *n* = 3–12.

**Figure 4 ijms-24-10939-f004:**
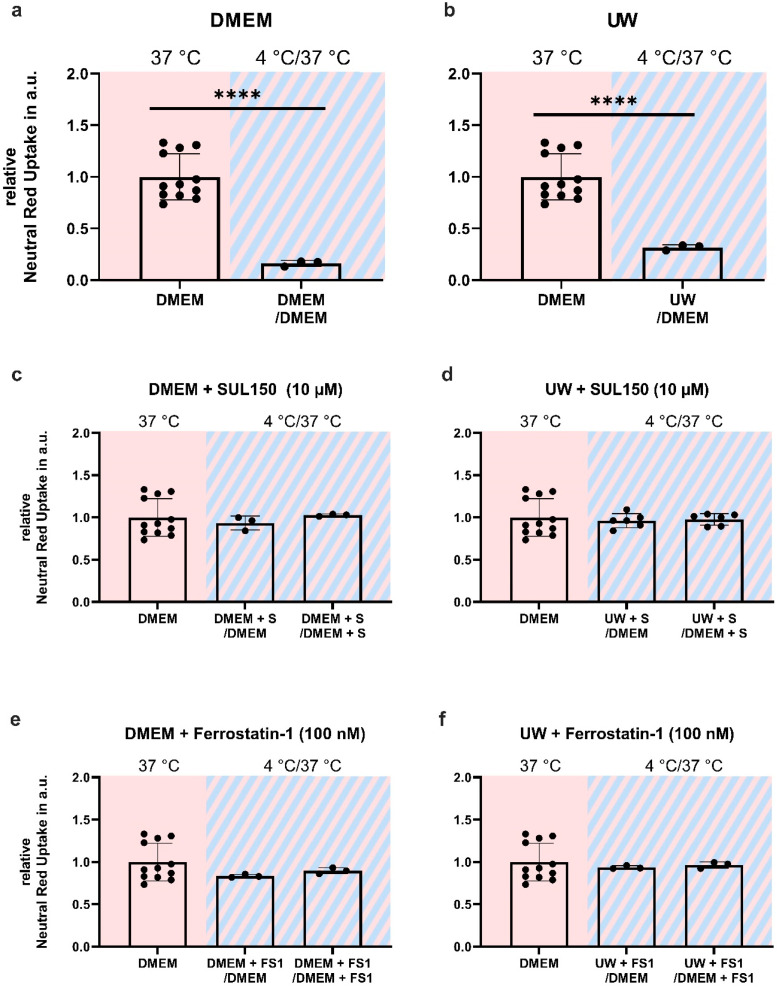
Cell Survival in Human Embryonic Kidney (HEK) 293 cells after 24 h of cooling and 1 h of rewarming (striped background) with SUL150 or Ferrostatin-1 supplementation in either DMEM or UW. (**a**,**b**) Cooling for 24 h and rewarming with DMSO vehicle in DMEM (**a**) and UW (**b**). (**c**,**d**) Cooling for 24 h and rewarming with SUL150 (S) in DMEM (**c**) and UW (**d**). (**d**,**e**) Cooling for 24 h with Ferrostatin-1 (F) in DMEM (**e**) and UW (**f**). All data are expressed relative to standard conditions at 37 °C and presented as means ± SD; **** = *p* < 0.0001; one-way ANOVA with Dunnett’s post hoc test; *n* = 3–12.

**Figure 5 ijms-24-10939-f005:**
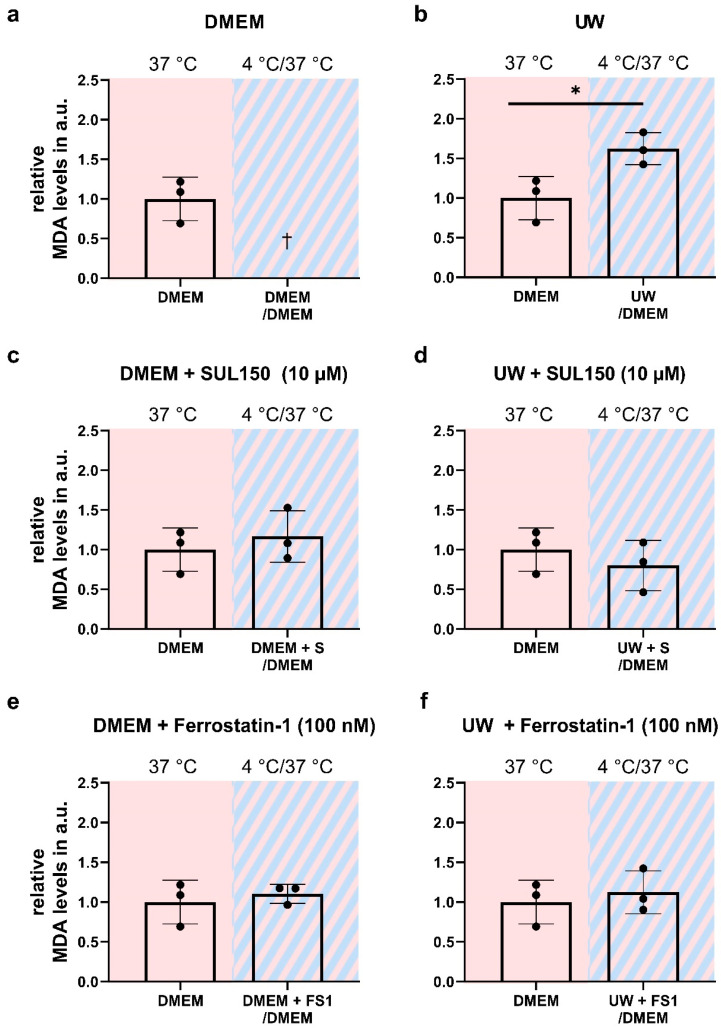
Lipid peroxidation (MDA) in University of Wisconsin Cold Storage Solution in Human Embryonic Kidney (HEK) 293 cells after 24 h cooling and 1 h rewarming (striped background). (**a**,**b**) Lipid peroxidation in DMEM (**a**) and UW (**b**). (**c**,**d**) Lipid peroxidation after SUL150 (S) treatment in DMEM (**c**) and UW (**d**). (**e**,**f**) Lipid peroxidation after Ferrostatin-1 (F) treatment in DMEM (**e**) and UW (**f**). All data are expressed relative to standard conditions at 37 °C and presented as means ± SD; one-way ANOVA with Dunnett’s post hoc test; * = *p* < 0.05; *n* = 3.

**Figure 6 ijms-24-10939-f006:**
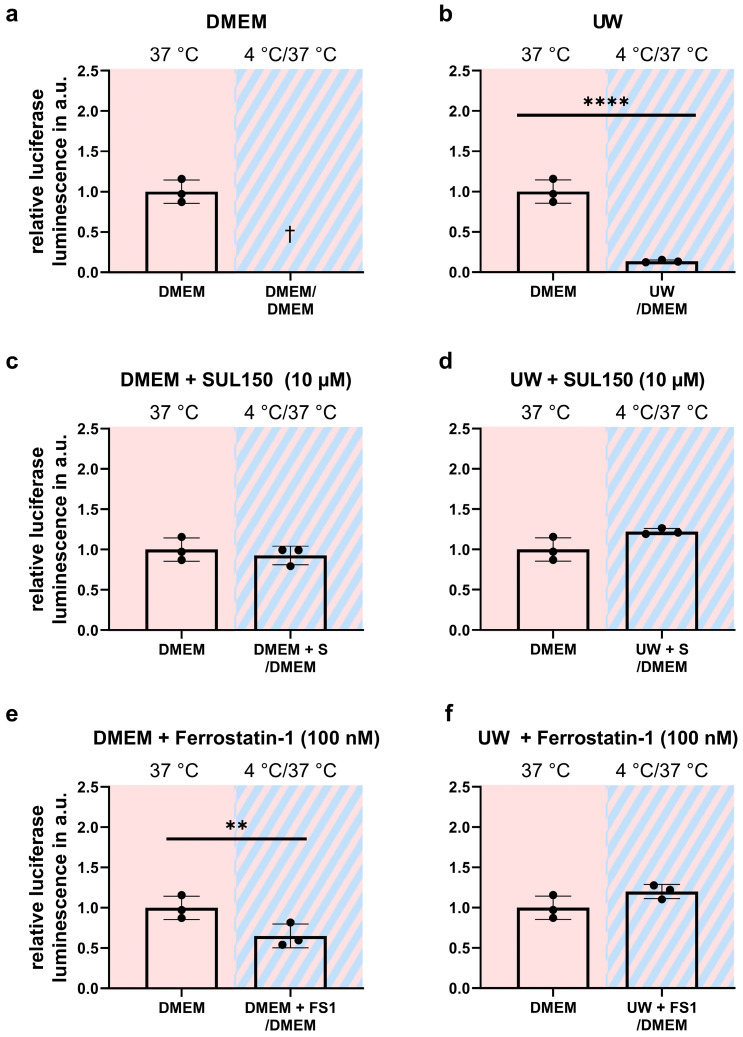
ATP luciferase activity in University of Wisconsin Cold Storage Solution in Human Embryonic Kidney (HEK) 293 cells after 24 h cooling and 1 h rewarming (striped background). (**a**,**b**) ATP in DMEM (**a**) and UW (**b**). (**c**,**d**) ATP after SUL150 (S) treatment in DMEM (**c**) and UW (**d**). (**e**,**f**) ATP peroxidation after Ferrostatin-1 (F) treatment in DMEM (**e**) and UW (**f**). All data are expressed relative to standard conditions at 37 °C and presented as means ± SD; one-way ANOVA with Dunnett’s post hoc test; ** = *p* < 0.01; **** = *p* < 0.0001; *n* = 3.

**Figure 7 ijms-24-10939-f007:**
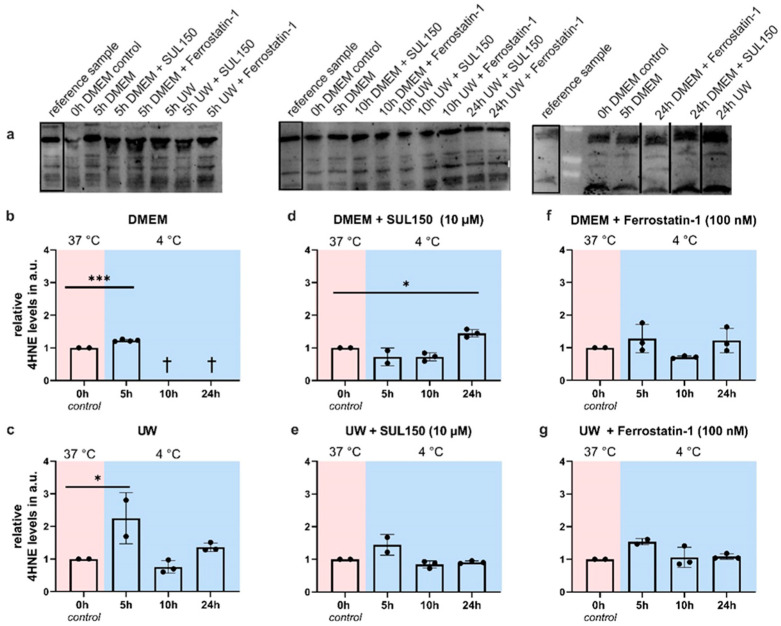
4HNE expression in Human Embryonic Kidney (HEK) 293 cells in a 5 to 24 h cooling model. (**a**) Typical examples of Western blots. (**b**) Cooling in DMEM. (**c**) Cooling in UW. (**d**) Cooling in DMEM + SUL150 (10 µM). (**e**) Cooling in UW + SUL150 (10 µM). (**f**) Cooling in DMEM + Ferrostatin-1 (100 nM). (**g**) Cooling in UW + Ferrostatin-1 (100 nM). Samples derived from the same experiment and blots were processed in parallel; the same reference sample was used on all blots and is highlighted with a rectangle. Vertical lines indicate the exclusion of lanes on the same blot. All data are expressed relative to standard conditions at 37 °C and presented as means ± SD; unpaired two-tailed *t*-test (B); one-way ANOVA with Dunnett’s post hoc test (C-G); * = *p* < 0.05; *** = *p* < 0.001; *n* = 2–5.

**Figure 8 ijms-24-10939-f008:**
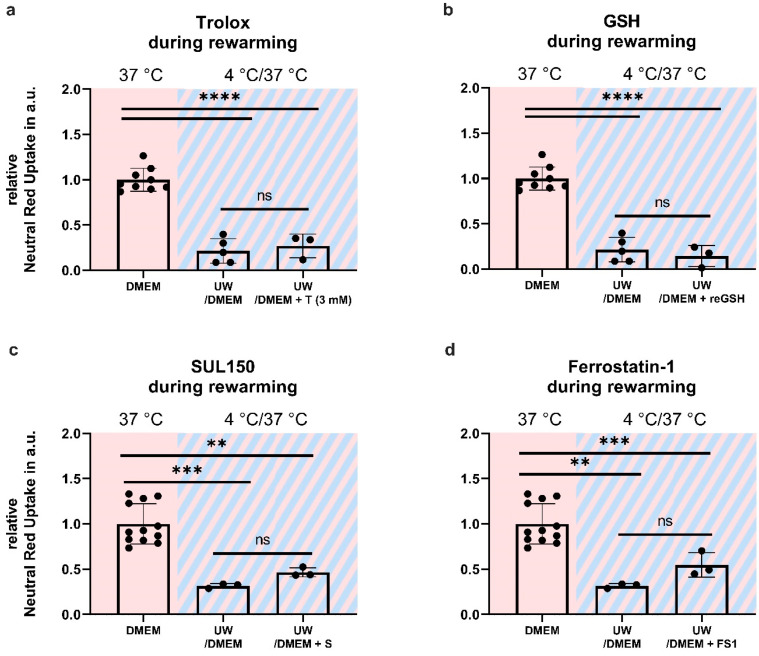
Cell survival in Human Embryonic Kidney (HEK) 293 cells in a 24 h cooling and 1 h rewarming model (striped background) with antioxidants added to the rewarming medium. (**a**) Addition of Trolox (T). (**b**) Addition of reduced GSH (reGSH). (**c**) Addition of SUL150 (S). (**d**) Addition of Ferrostatin-1 (F). All data are expressed relative to standard conditions at 37 °C and presented as means ± SD; one-way ANOVA with Šidák’s post hoc test; ** = *p* < 0.01; *** = *p* < 0.001; **** = *p* < 0.0001; *n* = 3–9.

**Figure 9 ijms-24-10939-f009:**
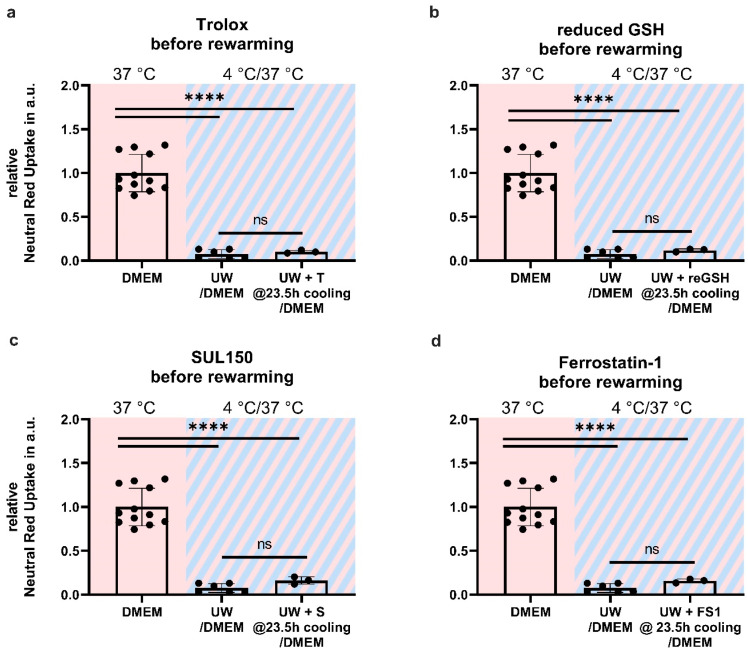
Cell survival in Human Embryonic Kidney (HEK) 293 cells in a 24 h cooling and 1 h rewarming model (striped background) with antioxidants added to the cooling medium 30 min before rewarming. (**a**) Addition of Trolox (T). (**b**) Addition of reduced GSH (reGSH). (**c**) Addition of SUL150 (S). (**d**) Addition of Ferrostatin-1 (F). All data are expressed relative to standard conditions at 37 °C and presented as means ± SD; one-way ANOVA with Šidák’s post hoc test; **** = *p* < 0.0001; *n* = 3–9.

**Figure 10 ijms-24-10939-f010:**
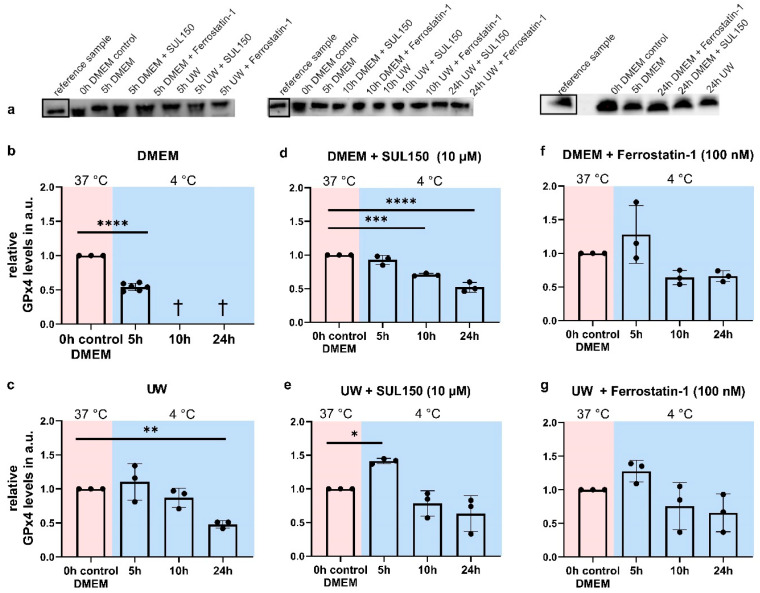
GPx4 expression in Human Embryonic Kidney (HEK) 293 cells in a 5 to 24 h cooling model. (**a**) Typical examples of Western blots. (**b**) Cooling in DMEM. (**c**) Cooling in UW. (**d**) Cooling in DMEM + SUL150 (10 µM). (**e**) Cooling in UW + SUL150 (10 µM). (**f**) Cooling in DMEM + Ferrostatin-1 (100 nM). (**g**) Cooling in UW + Ferrostatin-1 (100 nM). Samples derived from the same experiment and blots were processed in parallel; the same reference sample was used on all blots and is highlighted with a rectangle. Vertical lines indicate the exclusion of lanes on the same blot. All data are expressed relative to standard conditions at 37 °C and presented as means ± SD; unpaired two-tailed *t*-test (**b**); Grubbs outlier test (alpha = 0.05) removed one outlier in DMEM at 5 h; one-way ANOVA with Dunnett’s post hoc test (C-G); * = *p* < 0.05; ** = *p* < 0.01; *** = *p* < 0.001; **** = *p* < 0.0001; *n* = 2–5.

**Figure 11 ijms-24-10939-f011:**
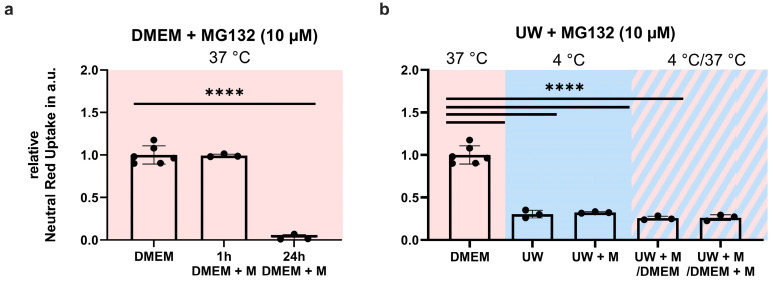
Proteasome blockage in Human Embryonic Kidney (HEK) 293 cells in a cooling (blue background) and rewarming (striped background) model. (**a**) MG132 (M) treatment under normothermic conditions. (**b**) Cell survival measured after 24 h cooling and rewarming using MG132 (proteasome inhibitor). All data are expressed relative to standard conditions at 37 °C and presented as means ± SD; one-way ANOVA with Šidák’s post hoc test; **** = *p* < 0.0001; *n* = 3–6.

**Table 1 ijms-24-10939-t001:** Used Chemicals.

Compound	Concentration	Manufacturer
SUL150	10 µM	Sulfateq B.V.
Ferrostatin-1	100 nM	
Lanthanum Chloride	50 µM	
Ru360 *	5 µM	
Trolox^®^	100 µM to 3 mM	Sigma-Aldrich
Reduced GSH	3 mM	
Oxidized GSH	3 mM	
DMSO	n.a.	
Z-VAD(OH)-FMK	50 µM	Selleck Chemicals
MG132	10 µM	
RSL3	1.44 and 2.88 µM	TargetMol
Erastin	5 and 10 µM	

All compounds were dissolved in DMSO, for which vehicle controls were performed. * Compounds were prepared after oxygen was removed through nitrogen displacement for 60 min.

## Data Availability

All data generated or analyzed during this study are included in this published article. The datasets of this article are accessible via the Appendix A.

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
