# Peer review of "Inhibition of Ferroptosis Enables Safe Rewarming of HEK293 Cells following Cooling in University of Wisconsin Cold Storage Solution"

_ijms, 2023, doi:10.3390/ijms241310939_

Round 1
Reviewer 1 Report
Prolonged cooling of cells causes cell death by apoptosis and ferroptosis which limits the usage of cold extended preservation. The authors studied the mechanism of cell death and efficacy of pharmacologic intervention in a cellular model of renal cold storage using the human embryonic kidney HEK293 cells and the use of the preservation solution University of Wisconsin (UW). They found that cell death from hypothermia and rewarming is mostly caused by ferroptosis rather than apoptosis. They suggest practical and low-cost solutions such as the addition of SUL150 and 459 ferrostatin-1 to the current UW to improve cold storage preservation tolerance of allografts and reduce complications such as ischemia.
The studies are interesting and well written and of practical usage for cold preservation of cells and organs. The below comments and experiments should be addressed:
The authors used the HEK293 cells as a model for their studies. These cells are tumorigenic. They should justify why they did not use normal-like kidney cells such as the Human Kidney-2 (HK-2) cells or UCL93?
The authors selected the neutral red assay as a sensitive and easily quantifiable indicator of cell viability due to the uptake of the dye neutral red which stains the lysosomes in viable cells. What is the cell death and viability in these cells? The authors should confirm these results by using trypan blue assays that measure viable and dead cells.
What are the levels of antioxidant enzymes (superoxide dismutase, catalase, glutathione peroxidase) and the antioxidant glutathione in control and treated HEK293 cells?
The authors use apoptosis inhibitors such as ZVAD. How do they know that these inhibitors worked? any controls that the inhibitors worked on the cells?
Data should be presented as the average +/- standard error of the mean instead of standard deviation.
How many were the western blots repeated and what is the protein loading and transfer control? Such as GAPDH, actin…
Minor comments:
Line 51: Q10 should be replaced by coenzyme Q10
Line 115: (partially) should be replaced by that is partially so the title reads “Hypothermia causes cell death in HEK293 cells that is partially ameliorated by UW.”
The authors should include more recent reviews of ferroptosis. There are more than 190 reviews of ferroptosis just published in 2023.
The authors need to comment how their in vitro studies can be used in kidney preservation.
Author Response
Please see attachement.

Reviewer 2 Report
The paper by Gartzke LP et al. showed that cell death from hypothermia and/or rewarming is caused by ferroptosis rather than apoptosis in HEK293 cell line. The data presented are interesting but there are some issues to be addressed:
Major points:
- The authors state that SUL150 prevents ferroptosis like Ferrostatin-1, but they do not present a direct demonstration of this. It is known that chromanols are potent inhibitors of ferroptosis, but the authors should cite the literature or show that SUL150 is capable of inhibiting cell death induced by, for example, ferroptosis inducers such as RSL3.
-Line 414-425 The authors report that for the atp assay (promega), the sample was boiled for 6 minutes. Boiling can degrade atp and is not usually expected for this type of determination. Also, this is an assay in which the amount of atp is usually detected by luminescence and not at 540nm. In Figure 6, the y-axis reads, "relative luciferase fluorescence"-you need to check that this is correct.
Minor points:
- Line 51: To help the reader read and understand the text, the authors should specify what “the Q10 effect” is.
- Figure 6: typos in y axis
- line 227: Is it not clear in which experiment the oxidized glutathione is used
- Line 248-250: the text refers to figure 10, not figure 7
none
Author Response
Please see attachement.

Round 2
Reviewer 1 Report
The authors have adequately addressed my revisions. The manuscript has improved upon these revisions.
Author Response
Thank you for reviewing and helping us to improve this article.
No document attached, as there was no further review points.
Reviewer 2 Report
I think the manuscript is now improved. However, I have a question: in Supplementary Figure 4, the ferroptosis-inducing effect of RSL3 and erastin is very mild. Why did the authors not test the efficacy of SUL150 on a stronger induction of ferroptosis, such as those presented in other figures in the manuscript?
Minor
- There are still some spelling errors in the figures (for example: buttom instead of bottom)
Minor editing of English language required
Author Response
Please, see attachment for changes.
